# Feasibility trial of a digital self-management intervention 'My Breathing Matters' to improve asthma-related quality of life for UK primary care patients with asthma

Ben Ainsworth [ID] ,[1,2] Kate Greenwell [ID] ,[3] Beth Stuart,[4] James Raftery,[4] Frances Mair,[5] Anne Bruton,[6] Lucy Yardley,[3,7] Mike Thomas[4]

For numbered affiliations see end of article.

**Correspondence to**
Dr Ben Ainsworth;
b.ainsworth@bath.ac.uk

## ABSTRACT

**Objective** To assess the feasibility of a randomised controlled trial (RCT) and acceptability of an asthma self-management digital intervention to improve asthma-specific quality of life in comparison with usual care.

**Design and setting** A two-arm feasibility RCT conducted across seven general practices in Wessex, UK.

**Participants** Primary care patients with asthma aged 18 years and over, with impaired asthma-specific quality of life and access to the internet.

**Interventions** 'My Breathing Matters' (MBM) is a digital asthma self-management intervention designed using theory, evidence and person-based approaches to provide tailored support for both pharmacological and non-pharmacological management of asthma symptoms.

**Outcomes** The primary outcome was the feasibility of the trial design, including recruitment, adherence and retention at follow-up (3 and 12 months). Secondary outcomes were the feasibility and effect sizes of specific trial measures including asthma-specific quality of life and asthma control.

**Results** Primary outcomes: 88 patients were recruited (target 80). At 3-month follow-up, two patients withdrew and six did not complete outcome measures. At 12 months, two withdrew and four did not complete outcome measures. 36/44 patients in the intervention group engaged with MBM (median of 4 logins, range 0–25, IQR 8). Consistent trends were observed to improvements in asthma-related patient-reported outcome measures.

**Conclusions** This study demonstrated the feasibility and acceptability of a definitive RCT that is required to determine the clinical and cost-effectiveness of a digital asthma self-management intervention.

**Trial registration number** ISRCTN15698435.

## INTRODUCTION

Asthma prevalence in the UK is among the highest in the world at nearly 6% of the UK adult population, comprising 5.4 million people, with the most managed in primary care. Although hospital admission and mortality rates for asthma improved from

### Strengths and limitations of this study

► This pragmatic randomised controlled feasibility trial examined 'My Breathing Matters' (MBM), a digital asthma self-management intervention that supported both pharmacological and non-pharmacological management of asthma symptoms.

► MBM was developed using theory, evidence and person-based approaches, and compared with standardised usual care (a booklet) with successful blinding and randomisation.

► Not all patients engaged with the intervention, and although numeric improvements in patient-reported asthma outcomes were larger in the active arm, improvements were observed in both arms.

1970 to 2000, these improvements have since stalled.[1] Surveys of asthma symptoms and health status impairment show that suboptimal control is common and that the majority of people with asthma in the UK frequently experience potentially avoidable symptoms and quality of life (QOL) impairment.[2]

Proactive self-management of asthma has been convincingly shown to improve clinical outcomes and have been advocated in guidelines for 25 year.[3] Guidelines are not always well implemented[4] and consequently some people with asthma do not receive evidence-based interventions that are known to impact positively on outcomes. Recent large-scale systematic reviews demonstrated that supported asthma self-management can reduce healthcare utilisation and increase asthma control, without increasing healthcare costs.[5 6] For example, people with asthma without a management plan are four times more likely to have an asthma attack needing emergency care in hospital, yet only 44% of people with asthma in the UK report having

BMJ

a self-management plan.[7] Self-management recommendations for asthma have also encompassed non-pharmacological strategies to improve control. These include lifestyle interventions, such as smoking cessation, allergen avoidance, weight reduction in those with obesity and breathing retraining interventions.[8]

Digital interventions (DIs) are increasingly recognised as a possible approach to achieve the aims of supporting chronic diseases, such as asthma. DIs can be convenient, easily accessed and may provide cost-effective tools by automating routine aspects of patient education, monitoring and support.[9] There is accumulating evidence that DIs are feasible and may be effective in the context of asthma. The Self-Management of Asthma Supported by Hospitals, Information and communication technology, Nurses and General practitioners (SMASHING) trial compared usual care with web-based educational resources, self-monitoring and automated feedback on medication titration, plus some group and email nurse support for patients with asthma. After 12 months, the intervention group had a better QOL and lung function and more symptom-free days, at no extra cost.[10] The Randomised trial of an Asthma Internet Self-management Intervention (RAISIN) pilot trial indicated that self-management interventions that included non-pharmacological (behavioural and psychological) components could be effective at improving QOL and asthma control, with improvements to 'reach' and response rate, by catering to patients with mild asthma but impaired QOL.[11] A recent systematic review and meta-analysis indicated that self-management DIs may be able to improve asthma control and reduce asthma-related QOL impairment[12]; however, there is limited evidence of benefit for other outcomes and larger confirmatory trials are required.

In the current randomised controlled feasibility trial, we developed and evaluated a digital self-management intervention, that incorporated pharmacological and non-pharmacological self-management support for adults in primary care with impaired asthma-specific QOL ('My Breathing Matters'; MBM), using evidence, theory and person-based approaches[13] and in line with Medical Research Council (MRC) guidance for developing and evaluating complex interventions.[14]

## Aim

The aim of the MBM study was to assess the feasibility of a trial to evaluate a DI in primary care to improve QOL and other clinical outcomes (such as asthma control, health resource use and lung function) of people with asthma, in comparison to usual care (with provision of standard patient information materials produced by the charity Asthma UK).

## Research objectives

1. To assess feasibility of trial procedures including recruitment strategy, eligibility criteria, consent, withdrawal, randomisation and blinding.

2. To assess feasibility of the MBM DI including usage and engagement.
3. To assess feasibility of data analysis, including data collection, data quality and management of trial data across trial endpoint measures to inform sample size calculations for a larger phase 3 randomised controlled trial (RCT).

## METHOD
### Design
We conducted a pragmatic feasibility RCT of the MBM DI in primary care.

### Setting
Eligible participants were identified from seven general practices from the Wessex, UK primary care research network to facilitate recruitment of people with varied socioeconomic status. To ensure we evaluated the intervention across a spread of socioeconomic deprivation, practices were purposively selected to be both rural (n=4) and urban (n=3), with mean practice deprivation index of 20.60% (SD 10.5); practice socioeconomic deprivation deciles=2, 4, 4, 5, 8, 10, 10, in which lower deciles indicate more deprivation.[15]

### Participants
Patients were included in the trial if they were aged 18 years or more, had physician-diagnosed asthma in their medical record, had received one or more antiasthma medication prescription in the previous 12 months, had impaired asthma-related health status (Asthma Quality of Life Questionnaire (AQLQ) score of less than 5.5 as assessing using a self-completed postal questionnaire), provided informed consent, were able to understand English and had access to the internet.

They were excluded from the trial if: (1) their general practitioner (GP) considered it inappropriate for them to take part (such as having an additional terminal condition), (2) they were attending a secondary care asthma clinic, or they were receiving either maintenance oral corticosteroids or injected biological treatments to control their asthma, (3) they were diagnosed with Chronic Obstructive Pulmonary Disease (COPD), (4) they had a household member already enrolled on the study or (5) they were judged by the research nurse to have 'unstable asthma' according to the clinical assessment and spirometry data at the baseline assessment (in which case they were referred back to their GP), or were diagnosed with 'difficult asthma' defined by the British Thoracic Society (BTS).

### Recruitment
Electronic searches of the computerised primary care medical record were conducted, and records screened by GP to remove ineligible participants. Invitation letters, study information sheets, consent forms screening questionnaires (Mini AQLQ[16]) and freepost return envelopes were posted to the participants who returned them if they

were interested in taking part. Patients who met screening criteria (AQLQ score of less than 5.5) were contacted by research team staff and attended a baseline appointment at their practice with a trained research nurse. Recruitment began in March 2017 and was completed in August 2017.

## Sample size

The target for this trial was to recruit 80 patients overall (40 per arm), in order to assess primary feasibility outcomes and to assess intervention engagement and acceptability.

## Randomisation and blinding

After completing outcome measures at their baseline appointment participants were randomised (block randomisation stratified by an average primary care AQLQ score (4.3) taken from a previous trial using the same inclusion criteria[8]). Information packs were given to participants after randomisation with instructions on signing up to MBM (if randomised to intervention group) or just usual care materials (if randomised to control). Research nurses conducting baseline appointments were blinded throughout the study until the final questionnaire that was only delivered to the intervention group.

## Interventions

### Intervention group: usual care with MBM and asthma UK booklet

Patients in the intervention group continued to receive usual care but were also given a code that allowed free unlimited access to MBM. MBM is a digital asthma self-management intervention that supports asthma self-management using both pharmacological and non-pharmacological approaches, developed using the Life-Guide Software[17] and described below according to the Template for Intervention Description and Replication (TIDieR) checklist.[18] A demonstration version of the intervention is available here: http://www.mybreathing-matters.co.uk. After signing up and completing quality-of-life-related self-monitoring questions, patients were offered tailored advice that directed them towards specific pharmacological or non-pharmacological sections of the online intervention. The pharmacological section provided information on different medication classes and inhalers, the use of personalised asthma action plans (PAAPs), encouraged medication adherence and gave information to facilitate and inform an effective asthma review with their GP. Pharmacological content was initially based on 'Living Well with Asthma', an asthma self-management intervention that previously demonstrated feasibility for self-management[11 19] and was developed in collaboration with people with asthma and with input from Asthma UK (a national asthma charity). This section was designed to answer common concerns about medication, incorporating a strategy described as 'the 4-week medication challenge' that encouraged participants to realise the benefits of adherence to regular medication, by self-monitoring their symptoms during 4 weeks of continuous inhaler use. The non-pharmacological

support included sections on a number of strategies to improve asthma control, such as breathing retraining, stress reduction and additional healthy lifestyle resources (physical activity, weight reduction, hand hygiene and smoking cessation). Optional nurse support was available by Asthma UK who provide a dedicated nurse helpline that was advertised through the intervention.

The intervention was developed using the person-based approach,[20] which places patients at the heart of the development process. Evidence from the primary mixed-methods research (such as Morrison *et al*[19]) and qualitative and quantitative reviews was used to develop guiding principles. A prototype intervention was piloted using 'think aloud' interview studies, in which patients with asthma used prototype versions of the MBM website and provided feedback on intervention acceptability and feasibility as they used it. In 46 interviews with 30 patients (purposively selected across a range of age and gender), the intervention was iteratively modified and updated to address patient feedback until participants indicated no further modification was required, confirming the intervention was as acceptable and engaging as possible for (see Yardley *et al*[21] for more details).

Due to the digital nature of the intervention, participants could engage with components of the intervention as much or as little as they wished. Tailored advice was offered according to participants' preference to find out more about pharmacological or non-pharmacological self-management techniques (patients selected a check box option of 'I'd like to find out more about how my asthma could be helped by (1) making the most of my asthma medicine or (2) 'non-medicine' ways to help my breathing.'), with automated reminders whenever patients had not accessed the intervention for several weeks, or when content was made available that they had not previously seen. The intervention was not modified during the study.

Intervention usage was monitored through digital usage metrics (reported below). Non-engagement with the website was not addressed, in line with the pragmatic nature of the feasibility study. Participants were sent one email and received one phone call, in which they were offered technical support if they had not logged onto the intervention at all for 1 month following their baseline appointment.

### Control group: usual care with asthma UK booklet

To provide 'good-quality' usual care to participants allocated to this arm, as well as usual care from their practice, participants were given an Asthma UK booklet 'Live Well with Asthma' at their baseline appointment. The booklet was created by a multidisciplinary team and expert patients, and aimed to provide essential information and advice to enable effective self-management to occur. It is available to anyone via the Asthma UK website.[22] The booklet was provided in a hard copy and provided information about asthma symptoms and triggers, medication adherence and usage techniques, PAAPs and support

from families. The booklet also advertised the Asthma UK support line. Booklet usage was not monitored.

## Outcome measures

In line with objectives, trial outcomes are reported below as (1) feasibility outcome measures, (2) intervention usage outcome measures and (3) trial endpoint measures to inform a larger trial.

### Feasibility outcome measures

Primary outcomes for the trial were descriptive, examining trial design and intervention feasibility and acceptability. These outcomes included patient recruitment, patient withdrawals and follow-up retention.

### Internet usage and engagement measures

Usage of the intervention included access to specific intervention components and frequency of engagement with individual components. These data were collected using the LifeGuide software.

### Endpoint measures

It is envisaged that the likely primary outcome measure or measures in a full trial would include validated asthma-specific patient-reported outcome measures evaluating symptom control and QOL, with additional secondary outcomes measuring health resource use, psychological measures and a health economic analysis. Data to generate hypotheses (and perform sample size calculations with which to test them) were collected in the following trial endpoints at baseline, 3 and 12 months:

► Asthma-specific QOL. Measured using the Mini-AQLQ[16], a 15-item 7-point scale in which higher scores represent higher QOL.
► Asthma control. Measured using the Asthma Control Questionnaire (ACQ[23]), a 7-item 7-point scale in which higher scores indicate worse asthma control.
► Health-related QOL. Measured using the EuroQol Five Dimensional Quality of Life (EQ-5D-5L),[24] in which participants select their functioning level across five dimensions (immobility, self-care, usual activities, pain and discomfort, and anxiety and depression) on a 5-point scale, with higher scores indicating greater problems with functioning.
► Health-related capability. Measured using the Icepop Capability measure for Adults (ICECAP-A[25]), in which participants select their capability across five dimensions (stability, enjoyment, achievement, attachment and autonomy) on a 4-point scale, with higher scores indicating better capability.
► Anxiety and depression. Measured using the Hospital Anxiety and Depression Scale (HADS[26]), a 12-item questionnaire in which higher scores on depression and anxiety subscales indicate higher anxiety across two subscales (anxiety and depression).
► Enablement. Measured using a modified version of the patient enablement instrument (PEI[27]) that has been validated in previous RCTs.[28] The modified

PEI is a 7-point scale consisting of six items, in which higher scores indicate more enablement.
► Patient satisfaction. Measured by asking patients whether they saw any benefits or disadvantages to using MBM, and whether they would recommend it to friends and family based on the National Health Service (NHS) friends and family test (FFT).[29]
► Patient burden was measured using a specifically developed questionnaire exploring time and costs via self-report based on Burden of Treatment Theory.[30] The questionnaire consisted of four questions with descriptive responses that explored whether new programmes were signed up to (such as gym membership, yoga/meditation) and the financial burden of doing so (see online supplementary appendix 1).

Physiological measures of lung function were taken at baseline and 12-month appointments: forced expiratory volume ($FEV_1$), ratio of $FEV_1$ to forced vital capacity ($FEV_1/FVC$) and peak expiratory flow rate (PEFR).

At 12-month follow-up, we also monitored health resource use GP consultations, Accident and Emergency Department (A&E) visits at, hospital admissions, asthma medication use and use of antibiotics for chest infections using GP practice patient notes.

Healthcare utilisation data were collected via retrospective notes review conducted by practice staff. Staff were provided with a template for reviewing data, and an instruction manual to ensure correct data were provided. Initial notes reviews were completed within two months from completion of the primary data collection. Nine patients (10% of the total patients in each practice) were also reviewed by research nurses to assess data quality.

## Patient and public involvement

Asthma UK was involved in the initial project proposal and supported the project throughout. Patient and public representatives (recruited with help from Asthma UK) participated in intervention development (providing feedback on prototype versions of the intervention, attending study management meetings, helping to develop trial materials and procedures and discussing responses to participant feedback). Asthma UK is involved in dissemination of this research and ongoing projects related to the research.

## Data analysis

Primary analysis of the study was a description of key feasibility outcomes including patient eligibility, recruitment rates, withdrawals, 3 and 12 months follow-up response rates and DI usage, as reported in the trial protocol (see online supplementary file 1).

Descriptive statistics were used to identify any floor or ceiling effects. For continuous measures, means, SD and 95% CIs were reported at baseline, 3 and 12 months for each group, as well as for the sample as whole.

Exploratory analysis explored group differences in continuous primary endpoint measures (AQLQ, ACQ, HADS and PEI) using linear regression models that

controlled for baseline values. Participants were analysed in the group to which they had been randomised and comprised complete cases only.

Proportions of patients achieving a minimal clinically important difference (MCID) were described for asthma QOL (the AQLQ MCID is 0.5[16]).

Sensitivity analysis explored missing data at 3 and 12 months.

Healthcare utilisation outcomes were explored using a negative binomial model of group count data.

Health economic analysis was descriptive, reporting estimates of cost and outcomes measures and baseline and follow-up. The completeness and suitability of EQ-5D-5L and ICECAP were compared as was the appropriateness of the resource use, and time and cost tools developed for the study.

Intervention engagement was descriptive.

## RESULTS

### Recruitment and retention

Six practices were initially recruited and after monitoring recruitment rates a seventh practice added. In this additional practice, only half of the list (randomly selected) were offered participation in the study to avoid over recruiting. Across the seven practices, 68 478 patients were assessed for eligibility with 3199 meeting initial eligibility criteria (asthma diagnosis, >1 asthma medicine prescription in last 12 months, screened by practice). 266 patients completed postal screening measures before the recruitment period finished, of whom 125 were eligible to take part (impaired asthma-related QOL, AQLQ score less than 5.5). Ninety patients responded to further contact. Two patients did not attend their baseline appointment leaving a final sample of 88 patients (intervention n=44, usual care n=44) who were recruited into the study (13.5 per practice) and were randomised over a

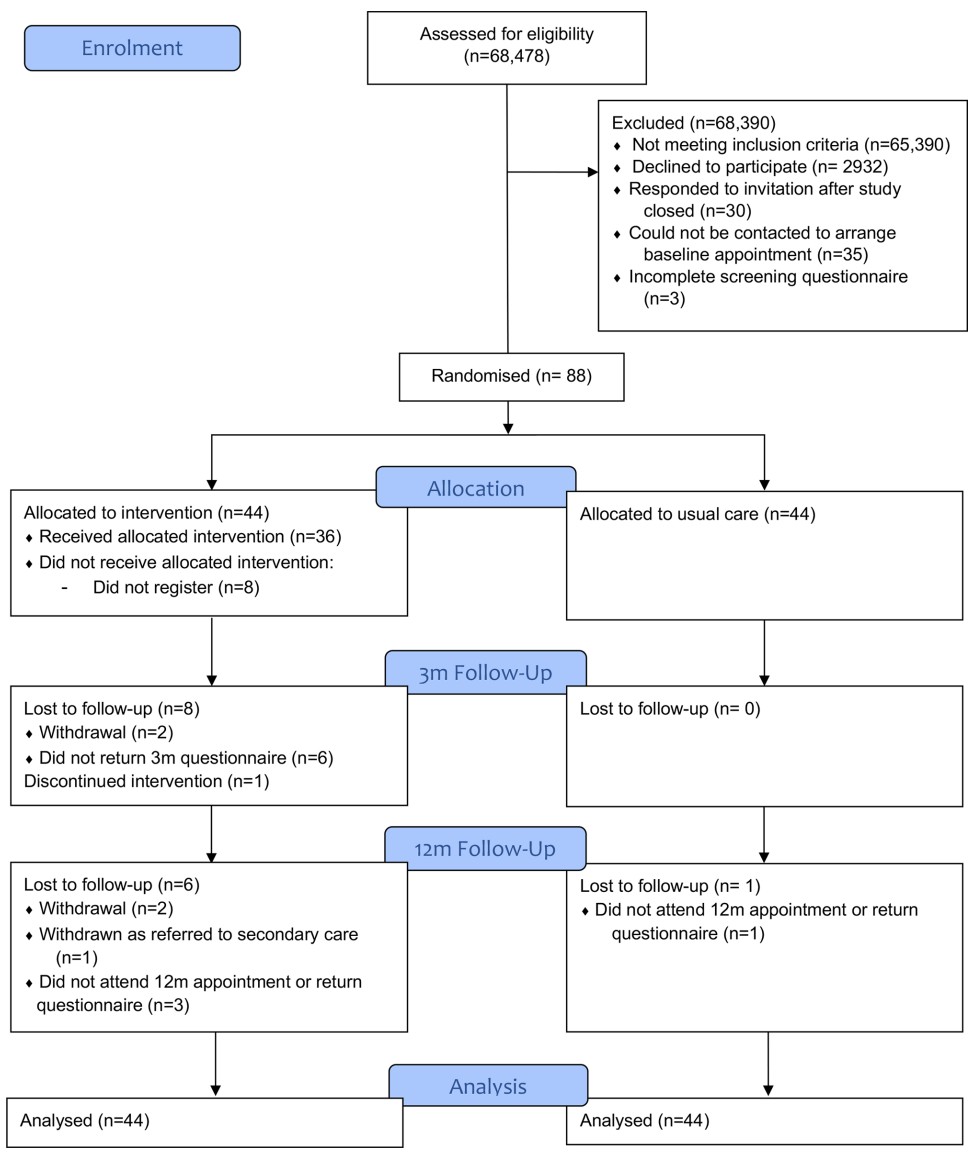

**Figure 1**  Study CONSORT diagram. CONSORT, Consolidated Standards of Reporting Trials.

5-month period. Figure 1 presents the study Consolidated Standards of Reporting Trials diagram.

During the study, two patients withdrew before 3-month follow-up and two before 12-month follow-up. All were in the intervention group. Patients withdrew for several reasons including lack of time (n=1), illness (n=1), death of family member (n=1) and lack of perceived benefit (n=1). Three of four participants who withdrew had used the intervention, one had not. One participant was withdrawn from the study prior to 12-month follow-up they were no longer eligible (ie, they were referred to secondary care).

Follow-up rates at 3 months were 91% (80/88; intervention: 36/44, control 44/44). Six (7%) patients did not complete 3-month follow-up measures but did not withdraw (all in intervention group).

At 12 months, 91% of participants provided primary outcome data by attending a follow-up appointment or returning a postal questionnaire (80/88; intervention: 37/45; control 43/43). 76% attended a baseline appointment and provided secondary clinical data (67/88). Four (5%) patients did not complete 12-month follow-up measures but did not withdraw (three in intervention group, one in usual care). None of these intervention participants had used the intervention. None of these patients responded to efforts to contact them by the study team.

## Patient characteristics

Demographics and baseline characteristics of participants are presented in table 1 and were reasonably well balanced between arms across all measures.

Table 1 compares those lost to follow-up to those who remained in the study at 3 months in a sensitivity analysis. Those lost to follow-up were slightly more likely to be female, have a higher body mass index (BMI), a longer time since diagnosis, a lower AQLQ score, a higher HADS-A and HADS-D score and to be from a more deprived postal code.

## Intervention usage and engagement

At 12-month follow-up, 36 (82%) patients in the intervention arm had engaged with the intervention (at least 1 log in). Patients logged in between 0 and 25 times to the intervention (median=4; IQR=8.25). Several patients also engaged with additional lifestyle modification interventions including improving hand hygiene (n=2), weight loss (n=3), improving physical activity (n=3) and getting support from friends and family (n=5).

After the study, participants in the intervention group were asked 'Do you think there were any benefits to using MBM?'. 12 of 36 (33%) reported 'quite a bit/a large amount of benefit', 19/36 (53%) reported 'some benefit' and 5/36 (14%) reported 'very little benefit'. 22 participants completed a free text box describing the advantages—benefits varied but included information provision (such as 'weight loss', 'dietary/exercise regimes'), medication adherence (such as asthma action

plans, improved medication adherence), provision of non-pharmacological treatments (such as breathing exercises and relaxation) and accessibility (such as 'access to information quickly'). This is reported in more detail in a separately published process analysis.

Participants were also asked 'Do you think there were any disadvantages to using MBM?'. 25 of 36 (69%) reported no disadvantages at all, 3 (8%) reported very few disadvantages, 8 (22%) reported some disadvantages and none reported quite a bit or a large amount of disadvantages. Thirteen participants completed a free-text box describing disadvantages, which included technical difficulties (such as not always accessible across different devices, difficulty logging in) and information specificity (such as not enough information, too many reminders, too few reminders). A final question asked how likely participants were to recommend MBM to friends or family. Sixteen participants (44%) were extremely likely to recommend it, 12 participants (33%) were likely, 7 (19%) were neither likely nor unlikely and 1 was extremely unlikely (3%).

## Trial endpoint measures

The full data of the trial endpoints are set out in table 2.

Both the intervention group and control group improved from baseline to 3 and 12-month follow-up, with numerically larger improvements in the asthma-related patient-reported outcomes measuring QOL and symptom control (AQLQ and AQC) at both time points; one or both these measures are anticipated to be the primary outcome of a subsequent fully powered study.

At the 3-month evaluation, patients in the intervention group who completed 3-month follow-up measures (n=36) had mean improvement in asthma-related QOL (AQLQ score) of 0.53 (95% CI 0.31 to 0.75), and in the control group of 0.52 (95% CI 0.30 to 0.74), with the between-group difference (controlling for baseline differences) the AQLQ being 0.06 higher (95% CI −0.22 to 0.35) in the intervention group, indicating better QOL. By 12 months, these figures were 0.35 (0.10 to 0.60) and 0.21 (−0.09 to 0.51), respectively, and the between-group difference had risen to 0.18 (95% CI −0.21 to 0.56) higher in the intervention group. In the ACQ analysis, at the 3-month analysis, the between-groups ACQ score was 0.14 lower (95% CI −0.41 to 0.13) in the intervention group, indicating better control, and at 12 months, it was 0.14 lower (95% CI −0.40 to 0.11). These findings indicate consistent trends to improvement in both asthma QOL and asthma control in the intervention group compared with the control. Full follow-up data are presented in table 2.

## Adverse events

Adverse events were reported by GPs and nurses who contacted the study team to report both adverse and serious adverse events. Nine adverse events were reported (intervention n=6, usual care n=3). These were assessed by research team clinicians and all were considered

**Table 1** Baseline demographic characteristics of study population per group.

| M (SD) | Overall sample (n=88) | Intervention group (n=44) | Control group (n=44) | Lost to follow-up (n=8) |
|---|---|---|---|---|
| Age | 56.6 (15.2) | 57.0 (14.2) | 56.3 (16.2) | 53.5 (12.11) |
| Female N (%) | 53.0 (60.2) | 27.0 (61.4) | 26.0 (59.1) | 6 (75) |
| BMI | 29.5 (6.1) | 28.9 (5.9) | 30.1 (6.3) | 32.7 (4.3) |
| Length of diagnosis | 24.0 (17.5) | 25.2 (17.2) | 22.8 (17.8) | 30 (18.9) |
| $FEV_1$ | 2.5 (0.8) | 2.6 (0.8) | 2.5 (0.8) | 2.40 (0.47) |
| % predicted $FEV_1$ | 92.3 (16.0) | 94.8 (16.0) | 89.8 (15.8) | 92.0 (12.9) |
| $FEV_1$/FVC | 76.6 (8.5) | 77.1 (8.0) | 76.1 (9.0) | 74.9 (4.1) |
| Peak flow | 421.2 (104.7) | 421.3 (108.3) | 421.1 (102.3) | 420.6 (83.8) |
| Ethnicity | | | | |
| White N (%) | 84 (95.5) | 42 (95.5) | 42 (95.5) | 7 (87.5) |
| Other N (%) | 4 (4.5) | 2 (4.5) | 2 (4.5) | 1 (12.5) |
| Smoking status | | | | |
| Current N (%) | 9 (10.2) | 7 (15.9) | 2 (4.5) | 2 (25.0) |
| Former N (%) | 29 (33.0) | 13 (29.5) | 16 (36.3) | 3 (37.5) |
| Never N (%) | 50 (56.8) | 24 (54.5) | 26 (59.1) | 3 (37.5) |
| Age left education | 18.5 (5.3) | 19.4 (7.0)* | 17.7 (2.7) | 20.4 (8.2) |
| 16 or under N (%) | 40 (46.5) | 18 (42.9) | 22 (50.0) | 4 (50.0) |
| 17–18 n (%) | 22 (25.6) | 9 (21.4) | 13 (29.5) | 1 (12.5) |
| Above 18 (%) | 24 (27.9) | 15 (35.7) | 9 (20.5) | 3 (37.5) |
| Index of Multiple Deprivation Mean Rank (median decile) | 17 192 (5.5) | 17 231 (6.5) | 17 212 (5) | 4505.5 (1.5) |
| AQLQ | 4.81 (1.01) | 4.85 (0.94) | 4.78 (1.09) | 4.26 (0.55) |
| ACQ | 1.45 (0.80) | 1.35 (0.66) | 1.56 (0.91) | 1.52 (0.73) |
| HADS-A | 6.60 (4.47) | 6.57 (3.87) | 6.64 (5.04) | 8.63 (3.9) |
| HADS-D | 3.89 (3.57) | 3.39 (3.07) | 4.39 (3.99) | 4.75 (4.4) |
| EQ-5D-5L | 0.83 (0.19) | 0.86 (0.15) | 0.81 (0.22) | 4 (50.0) |
| EQ-5D-VAS | 71.5 (18.2) | 70.0 (19.3) | 73.0 (17.2) | 1 (12.5) |
| ICECAP-A | 0.87 (0.18) | 0.89 (0.12) | 0.88 (0.16) | 3 (37.5) |
| PEI | 2.52 (1.23) | 2.44 (1.09) | 2.60 (1.37) | 2.73 (1.0) |
| MARS-A | 4.70 (1.05) | 4.80 (0.90) | 4.60 (1.20) | 4.3 (0.8) |

*Percentages are reported from 42 participants as 2 participants in the intervention group did not complete these data.
ACQ, Asthma Control Questionnaire; AQLQ, Asthma Quality of Life Questionnaire; BMI, body mass index; EQ-5D-5L, EuroQol Five Dimensional Quality of Life; EQ-5D-VAS, EuroQol-5D Visual Analogue Scale; $FEV_1$, forced expiratory volume; FVC, forced vital capacity; HADS-A, Hospital Anxiety and Depression Scale-Anxiety; ICECAP-A, Icepop Capability measure for Adults; MARS-A, Medication Adherence Rating Scale for Asthma; PEI, patient enablement instrument.

unlikely to be related to the study. Three were related to participant asthma (asthma exacerbation not leading to hospital admission, upper respiratory tract infection, sinusitis).

Three serious adverse events were reported (intervention n=2, usual care n=1). These were considered unlikely to be related to the study and the condition (atrial fibrillation, open distal radius fracture, cardioversion).

### Healthcare utilisation outcomes

Data were collected from the retrospective notes reviews (conducted by practice nurses) from 83 participants, reported in table 3. Data were collected from 84 practices for seven participants, with four participants from one practice incomplete. The data quality check and subsequent examination by research team clinicians (MT) found that reviews completed by the practice nurses varied substantially in quality with varied levels of detail, and the quality of data achieved in this way was insufficient for a health economic analysis.

Comparisons between group count data were reported using a negative binomial model but given the issues with the reliability of the data, should be interpreted

**Table 2** Three and 12 months follow-up data (corrected for baseline differences)

| Measure | Intervention group (n=36) | | | Control group (n=44) | | | Difference between the intervention and control group controlling for baseline (95% CI) |
|---|---|---|---|---|---|---|---|
| | M (SD) | %>MCID* improvement | % items complete | M (SD) | %>MCID* improvement | % items complete | |
| 3 months | | | | | | | |
| AQLQ | 5.51 (0.85) | 47.2 | 82 | 5.30 (1.07) | 47.7 | 100 | 0.06 (−0.22 to 0.35) |
| ACQ | 0.98 (0.65) | | 82 | 1.28 (0.87) | | 100 | −0.14 (−0.41 to 0.13) |
| HADS-A | 6.75 (3.85) | | 82 | 7.07 (5.48) | | 100 | −0.04 (−0.18 to 0.11) |
| HADS-D | 3.75 (2.82) | | 82 | 4.66 (4.99) | | 100 | −0.02 (−0.16 to 0.13) |
| PEI | 2.71 (1.09) | | 82 | 2.90 (1.14) | | 100 | −0.12 (−0.59 to 0.35) |
| MARS-A | 4.23 (0.70) | | 80 | 4.05 (0.74) | | 100 | 0.04 (−0.25 to 0.3) |
| EQ-5D-5L | 0.82 (0.19) | | 82 | 0.83 (0.20) | | 100 | – |
| ICECAP-A | 0.87 (0.12) | | 82 | 0.84 (0.19) | | 100 | – |
| 12 months | | | | | | | |
| AQLQ | 5.29 (0.98) | 38.9 | 82 | 5.00 (1.25) | 39.5 | 98 | 0.18 (−0.21 to 0.56) |
| ACQ | 1.00 (0.59) | | 82 | 1.26 (0.69) | | 98 | −0.14 (−0.40 to 0.11) |
| HADS-A | 7.78 (3.94) | | 84 | 6.63 (4.91) | | 98 | 0.99 (0.16 to 2.15) |
| HADS-D | 3.81 (3.54) | | 84 | 4.19 (4.17) | | 98 | 0.22 (−0.97 to 1.41) |
| PEI | 2.46 (1.03) | | 84 | 2.61 (1.28) | | 98 | −0.09 (−0.54 to 0.37) |
| MARS-A | 4.37 (0.81) | | 82 | 4.29 (0.85) | | 98 | −0.09 (−0.43 to 0.25) |
| EQ-5D-5L | 0.83 (0.21) | | 82 | 0.80 (0.23) | | 98 | – |
| ICECAP-A | 0.86 (0.13) | | 82 | 0.84 (0.20) | | 98 | – |
| $FEV_1$ (litres) | 2.75 (0.75) | | 57 | 2.43 (0.74) | | 80 | 0.03 (−0.05 to 0.10) |
| $FEV_1$/FVC | 78.8 (6.58) | | 57 | 76.3 (9.29) | | 80 | 2.20 (−0.13 to 4.27) |
| % predicted $FEV_1$ | 100.1 (14.8) | | 57 | 92.4 (13.8) | | 80 | 1.77 (−1.72 to 5.25) |
| Peak flow | 450 (105) | | 57 | 417 (102) | | 80 | 15.29 (−6.27 to 36.86) |
| BMI | 28.7 (6.17) | | 64 | 31.1 (6.51) | | 86 | −0.11 (−0.89 to 0.68) |

*There was no difference in the number of patients who showed MCID improvement at 3 months (AQLQ, >0.5) across groups (47.2% in the intervention group compared with 47.7% in the control group). The same was true at 12 months (38.9% compared with 39.5%).
ACQ, Asthma Control Questionnaire; AQLQ, Asthma Quality of Life Questionnaire; BMI, body mass index; EQ-5D-5L, EuroQol Five Dimensional Quality of Life; $FEV_1$, forced expiratory volume; FVC, forced vital capacity; HADS-A, Hospital Anxiety and Depression Scale-Anxiety; ICECAP-A, Icepop Capability measure for Adults; MARS-A, Medication Adherence Rating Scale for Asthma; MCID, minimal clinically important difference; PEI, patient enablement instrument.

**Table 3** Data on asthma-related medication use (during the study period)

| Healthcare utilisation (N, IQR) Mean (SD) | Intervention group | | Control group | |
|---|---|---|---|---|
| | 12 months before study period | 12 months after study period | 12 months before study period | 12 months after study period |
| SABA prescriptions | 3 (2,6) 3.92 (3.48) | 3 (1,6) 4.00 (3.72) | 3 (2,5) 4.0 (3.89) | 4 (2,6) 4.39 (3.81) |
| ICS prescriptions | 5 (2,11) 6.72 (4.92) | 4 (3,10) 6.15 (4.21) | 6 (4,10) 7.41 (5.45) | 6 (4,10) 7.34 (5.37) |
| Oral steroids prescriptions | 0 (0,0) 0.31 (0.80) | 0 (0,0) 0.36 (0.94) | 0 (0,0) 0.43 (1.07) | 0 (0,0) 0.23 (0.71) |
| Antibiotic prescriptions | 0 (0,0) 0.33 (0.87) | 0 (0,0) 0.28 (0.60) | 0 (0,0) 0.52 (1.45) | 0 (0,1) 0.16 (0.48) |

ICS, *Inhaled corticosteroids*; SABA, Short-Acting Beta Agonist.

cautiously. The prescription rate was approximately 8% higher for both SABA incidence rate ratio (IRR 1.08, 95% CI 0.82 to 1.43) and ICS (IRR 1.08, 95% CI 0.86 to 1.35) in the control group compared with the intervention group. Both groups had a low number of prescriptions for oral steroids, oral steroids and antibiotics, with only 15 prescriptions in total for either of these medications, making between-group comparisons unreliable.

Due to unreliability of data, frequency of GP consultations, A&E admissions and hospitalisations have not been reported.

### Health economic outcomes

Both EQ-5D-5L and ICECAP-A had the same completion rates as other secondary measures completed at follow-up (see table 2).

Patients reported several programmes across both groups including gym, walking, yoga, sewing, language courses, physio and signing (see online supplementary file 2). There were no substantial differences in terms of numbers or costs although the sample size was small.

### DISCUSSION

In line with our main research objectives, findings from our randomised controlled feasibility trial demonstrate that a full-size confirmatory trial to confirm effectiveness of MBM, a digital self-management intervention for adults in primary care with asthma is likely to be feasible and acceptable. Our trial procedures, intervention usage and data management were all feasible. There were also trends to improved asthma control and QOL in our underpowered sample, so supporting the need for a definitive fully powered study. Our recruitment procedures recruited a specific patient sample (those impaired asthma-specific QOL) to target from a range of urban and rural practices.

Our sample varied in age with a relatively high mean (56 years) indicating that our DI can provide benefit to older adults. Both male and female adults were well represented in our trial. A notable proportion of our sample was obese (41%), in line with previous findings.[11] Given that obesity is a risk factor for asthma, a larger trial could further improve effectiveness by providing more specific behavioural content for obese adults with asthma (such as tailored content to increase motivation to use weight loss-related lifestyle components in obese patients). Our sample was also predominantly white. Under-representation of minority ethnic groups in medical research in the UK is an ongoing issue[12] and should be addressed in recruitment procedures in the full trial.

The feasibility of a full trial is supported by the effective completion of trial procedures. All patients who completed baseline measures were randomised. Completion of measures was good at both follow-up points (3 months via post and 12 months at participants' practice). Where participants were not able to attend a follow-up appointment at practices, they were satisfactorily followed up via post or telephone for main trial measures. Eight

patients were lost to follow-up (four withdrew and four no longer responded to attempts to contact them). Notably, all eight patients lost to follow-up were in the intervention group. It is possible that patients in the control group were more likely to maintain contact as they were only able to access the intervention on completion of 12-month follow-up measures. Although lost to follow-up is low, it is important to consider whether that loss is differential. Those lost to follow-up were more likely to be more socioeconomically deprived, female, have a higher BMI, a longer time since diagnosis and a higher HADS-A and HADS-D score. It is possible that these patients would benefit from using MBM more than most, and therefore, we have proposed several ways to further increase trial efficacy. Automatic email intervention registration at baseline (patients cannot attend baseline appointment without enrolling on the intervention) would increase initial engagement and engagement with trial procedures throughout duration of study. Online questionnaire completion during screening process would (1) screen patients who are unable to interact with online trial/intervention and therefore unable to benefit from the intervention (feasibility trial estimate=2%), and (2) streamline baseline/follow-up procedures.

Both health economic outcomes had high completion rates but did not suggest substantive change, similar to EQ-5D measures in previous non-pharmacological self-management trials (such as Bruton et al[8]). It is possible that an alternative measure such as the Short Form 12-item Survey[31] in which participants consider the previous 2 weeks (whereas in the EQ-5D they consider the immediate present) may be better suited to measure small yet valuable changes in well-being over a full trial. Our detailed mixed-methods process analysis explored issues of trial acceptability in more detail, and will be reported in a subsequent paper. A full trial of this non-pharmacological intervention should accurately capture 'non-medical' costs (such as gym membership) that are likely to impact disease-specific QOL, as well as medical costs that would be affected by changes in healthcare utilisation.

Healthcare utilisation data were collected by practice nurses whose main role was to provide usual clinical care at the practices, using a manual to guide data collection, rather than by trained research nurses, and our quality check demonstrated that the data collection process used was unreliable some centres. We conclude that in a full subsequent study, these data should be collected from the medical record by a trained member of the study team (such as a trained research nurse), as has been successfully used in previous studies.[8]

Engagement with the intervention was slightly increased compared with a previous similar digital asthma self-management intervention[19] at initial sign up (82% vs 76%) as well as maintaining a higher number of ongoing engagement throughout the follow-up period (median 3 additional log ins vs 1), although our study used a broad primary care population while the RAISIN protocol primarily recruited

from areas of high deprivation. This finding demonstrates that the use of the person-based approach to develop the intervention resulted in an intervention that was acceptable and engaging to patients, even using a pragmatic methodology in which patients self-registered at home instead of being registered by a GP during their baseline appointment. Participants accessed both pharmacological and non-pharmacological self-management content. We further explored the acceptability of the intervention to people with asthma in a mixed-methods process analysis which will be published separately.

Estimates of effect size demonstrated that participants who received the intervention and completed follow-up measures showed improved and clinically relevant QOL and asthma control. The order of magnitude of the mean between-group improvements in the patient reported measures of control (ACQ) and asthma-related QOL (AQLQ), although not statistically significant with the sample size of this feasibility study, was comparable to that reported in controlled studies of pharmacological[32] and non-pharmacological[8] interventions in asthma, and so justify a confirmatory study with a fully powered sample.

There was no suggestion of an effect on physiological measures of lung function. These results are in line with previous studies of behavioural self-management interventions in primary care adults with asthma (such as BREATHE, RAISIN), and demonstrate the importance of interventions targeting outcomes that incorporate elements of functional well-being (disease-specific QOL, subjective symptoms), rather than solely focusing on objective, physiological measures that are not correlated with quality of life.

The effectiveness of our intervention could be further increased according to findings from our process evaluation. This analysis, which will be reported separately, broadly agrees with previous research[19] in finding that many patients consider their asthma to be 'well controlled' despite having important levels of symptoms and QOL impairment on validated questionnaire. This implies that many people had become accustomed to their ongoing symptoms and had altered their life to try to reduce their impact, using denial as a coping mechanism. As a consequence, the means of appropriately targeting and framing self-management interventions should be carefully considered in future work, focusing on maintaining good health rather than improving poor health. Some of our findings (such as the association between QOL improvement and ongoing intervention engagement) demonstrate that framing content as positive and not focusing on illness—for example 'How to keep your breathing healthy' rather than 'How to reduce asthma symptoms' may lead to an acceptable, engaging intervention that benefits this patient group.

There were some limitations to this small feasibility study. Although our researchers and statisticians were blind to group allocation, patients would have known that they were allocated to the intervention rather than the usual care control. This is common in complex behavioural interventions. Furthermore, although we endeavoured to recruit participants across a broad demographic range, the reach

of our intervention could be improved. While the reach of DIs improves as digital literacy increases nationally, care must be taken to ensure that 'digital transformation' of NHS services does not entrench healthcare inequality, by facilitating a 'digital divide' that fails to provide adequate health and social care to those who do not have the digital skills to benefit.

## CONCLUSION

Our findings demonstrate the feasibility of a new digital self-management intervention for asthma (MBM). Using the person-based approach to intervention development means that MBM is both acceptable and engaging for adults with asthma in primary care. MBM reflects the varied experiences of people with asthma, by including both non-pharmacological and pharmacological components. Our data support the feasibility of moving towards a fully powered RCT, with only minor modifications to some trial procedures required.

**Author affiliations**
[1]Department of Psychology, University of Bath, Bath, UK
[2]NIHR Respiratory Biomedical Research Unit, University Hospital Southampton NHS Foundation Trust, Southampton, UK
[3]Psychology, Faculty of Environmental and Life Sciences, University of Southampton, Southampton, UK
[4]Primary Care and Population Sciences, Faculty of Medicine, University of Southampton, Southampton, UK
[5]Institute of Health and Wellbeing, University of Glasgow, Glasgow, UK
[6]School of Health Sciences, Faculty of Environmental and Life Sciences, University of Southampton, Southampton, UK
[7]School of Psychological Sciences, University of Bristol, Bristol, UK

**Acknowledgements** The authors would like to acknowledge the contributions of all participants and patient and public representatives in the development of the My Breathing Matters intervention development and feasibility evaluation, and Deborah Morrison who lead the development of 'Living Well With Asthma' resource that guided development of My Breathing Matters. The study was supported by the National Institute of Health Clinical Research Network (NIHR CRN).

**Contributors** LY, MT and AB conceived the idea for the study. LY, MT, JR, FM, BS and AB secured funding for the study. BA, LY, AB, BS and MT developed the intervention and designed the trial with input from Asthma UK and FM. BA and KG managed the trial on a day-to-day basis with support from LY and MT. BS and JR planned and carried out the statistical analysis. BA drafted the manuscript with assistance and final approval from all authors. LY is the guardian of the data.

**Funding** This article presents independent research funded by the National Institute for Health Research (NIHR) under its Programme Grants for Applied Research (PGfAR) Programme (Grant Reference Number RP-PG-1211-20001). My Breathing Matters was developed using LifeGuide software, which was partly funded by the NIHR Southampton Biomedical Research Centre (BRC). LY is an NIHR Senior Investigator and during the study BA was supported by an NIHR School of Primary Care (SPCR) Fellowship. AB was funded by the National Institute for Health Research (NIHR) Senior Research Fellowship (SRF-2012-05-120).

**Disclaimer** The views expressed are those of the authors and not necessarily those of the NIHR or the Department of Health and Social Care.

**Competing interests** Neither MT nor any member of his close family has any shares in pharmaceutical companies. In the last 3 years, he has received speaker's honoraria for speaking at sponsored meetings or satellite symposia at conferences from the following companies marketing respiratory and allergy products: GSK, Novartis. He has received honoraria for attending advisory panels with; Boehringer Ingleheim, GSK, Novartis. He is a recent a member of the BTS SIGN Asthma

guideline steering group and the NICE Asthma Diagnosis and Monitoring guideline development group. BA, KG, JR, BS, LY, FM and AB have no competing interests.

**Patient consent for publication**  Not required.

**Ethics approval**  The study was approved by NHS South Central—Berkshire Research Ethics Committee, 16/01/2017, ref: 16/SC/0614.

**Provenance and peer review**  Not commissioned; externally peer reviewed.

**Data availability statement**  Data are available on reasonable request.

**ORCID iDs**
Ben Ainsworth http://orcid.org/0000-0002-5098-1092
Kate Greenwell http://orcid.org/0000-0002-3662-1488

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
