## [Reviewer comments · BMJ Open]

ARTICLE DETAILS

TITLE (PROVISIONAL)	A feasibility trial of a digital self-management intervention 'My Breathing Matters' to improve asthma-related quality of life for UK primary care patients with asthma
AUTHORS	Ainsworth, Ben; Greenwell, Kate; Stuart, Beth; Raftery, James; Mair, Frances; Bruton, Anne; Yardley, Lucy; Thomas, Mike

VERSION 1 – REVIEW

REVIEWER	Elida Zairina Faculty of Pharmacy, Universitas Airlangga, INDONESIA
REVIEW RETURNED	12-Jul-2019

GENERAL COMMENTS	Thank you sincerely for the opportunity to review this manuscript. This is an interesting and important area of study to see the feasibility of asthma self-management supporting by digital intervention. The study has also published the protocol of the study trial. Please find bellows are my comments to improve the manuscript. Abstract: Objective / Aim: ...improve asthma specific QoL.. Is this study specifically designed to see the feasibility of DI asthma self-management to improve asthma related Specific QoL or other clinical outcomes (line 40 page 5), please confirm Introduction: The cited research in the introduction explained about the evaluation of other trial to improve asthma outcomes, I am not sure if this study also explored the improvement of asthma outcomes in the trial group? Methods: Setting: Are all those General Practices similar one to another? i.c. has a similar characteristics of asthma patients, in the area of remote or urban? Outcomes measures: It doesn't clear what are the instruments used for measuring Asthma –related QoL? Was it mini AQLQ or AQLQ? Page 6 line 33 mentioned AQLQ, Line 30 page 10, mentioned mini AQLQ (32 items), line 30 page 12 please confirm since Juniper's mini AQLQ had 15 items questions (short version of 32 items in AQLQ) Regarding the feasibility outcomes, I wonder why outcomes regarding the acceptability of using the DI MBM was measured. Such as whether it designed to be used by all ages of asthmatic patient easily? Results: Page 13 line 54
--

	Primary outcomes intervention 37/45 and control 43/43 – the less number on the intervention group completed the study? Page 15 line 6 Have any statistical analysis done to see both groups are well-balance and similar? Page 16 line 45 Are this separately process analysis already published? What it has to be separated? if it could be included in this study. Page 17 line 21 (Not statistically significant)...not clear what it means. Is this study expected to be significant statistically? The larger improvement just showed descriptively. I am not sure if the study is power enough to be shown statistically significant. Discussion: Page 22 line 44: Are both study are similar to compare regarding the intervention? Sample size? And the outcomes measures? If not, stated that the engagement with the intervention was increased is not fully correct. Conclusion: I have not seen clearly which part of the study that saying the MBM is feasible, since the analysis was done descriptively, and there were no significant difference in outcomes measured between Intervention and control group for each outcome.
--	---

REVIEWER	Demetris Lamnisos Department of Health Sciences, European University Cyprus
REVIEW RETURNED	18-Jul-2019

GENERAL COMMENTS	Review on the statistical methods and analysis The aim of this study was to assess the feasibility of a trial to evaluate a digital intervention in primary care to improve clinical outcomes of people with asthma, in comparison to usual care. As this is the aim of the study, then any statistical method should be exploratory and it has low power to make any inference about main effects. Below are few comments on the statistical methods and the reporting:  1. Page 2 - line 46: report Interquartile Range (IQR) as well. 2. Page 3 - line 23: A powered study has an ability to identify a true difference between the groups if such difference exist and does not always results in statistical significance results. Therefore remove the sentence “and the study was not powered to show statistical significance” and add if you like the sentence “and the sample size of the study was not determined from a power analysis”. 3. Page 12 – line 34: All measures were considered continuous. If this is the usual practice and the suggestions of the creators of those measures then it is fine to consider them continuous and apply linear regression models. Otherwise, adjust according to the suggestions of the creators of those measures. 4. Page 12 – line 34: Report all the independent variables included in the multiple linear regression model. 5. Page 12 – line 35: It is suggested to use a multilevel repeated measurement model. The highest level is the practices (7 general practices) and the other level is repeated measurements on the individuals over time. 6. Page 12 – line 41: Define the minimal clinically important difference. 7. Page 17 – line 41: Correct the 95% Confidence Interval. It should be (-0.22, 0.35).
---

	8. Page 17 – line 45: The same as above. 9. Table 2 – last column: Are those the results of the difference between the means of the intervention and control group? If they are then the AQLQ is 0.06 lower in the intervention group. Please correct accordingly the Table and the statements in the manuscript. 10. Page 20 – line 6: Mention the negative binomial model in the Data Analysis section. 11. Page 20: Report sensitivity analysis results as you mention it in the Data Analysis. 12. Page 23 – line 21: Power analysis methods should justify the power of the study. Therefore remove the sentence “and so justify a fully powered study”.
--	--

VERSION 1 – AUTHOR RESPONSE

Comments from Reviewer 1

1. Thank you sincerely for the opportunity to review this manuscript. This is an interesting and important area of study to see the feasibility of asthma self-management supporting by digital intervention. The study has also published the protocol of the study trial. Please find bellows are my comments to improve the manuscript.

We thank the reviewers for their positive comments.

2. Abstract :Objective / Aim: “...improve asthma specific QoL..”. Is this study specifically designed to see the feasibility of DI asthma self-management to improve asthma related Specific QoL or other clinical outcomes (line 40 page 5), please confirm.

We can confirm that the main outcome of the intervention was to improve asthma specific quality of life. We have clarified the aim of the study (page 5, line 40) to now read: “The aim of the MBM study was to assess the feasibility of a trial to evaluate a digital intervention in primary care to improve quality of life and other clinical outcomes (such as asthma control, health resource use, lung function)”

3. Introduction: The cited research in the introduction explained about the evaluation of other trial to improve asthma outcomes, I am not sure if this study also explored the improvement of asthma outcomes in the trial group?

The ERS-ATS Task force on asthma outcomes in clinical trials (Reddel et al., 2009, Am J Respir Crit Care) states that no single outcome measure can encompass the complexity of asthma, and that a range of outcome measures are appropriate, including measures of physiology, biomarkers of inflammation, symptom and quality of life measures. In line with this, we have used a number of these recommended metrics in our study. The task force recommends that the primary outcome in asthma studies will vary between studies depending on the focus of the study. Asthma-specific quality of life measures the impact of asthma on a patient’s life, and we feel this is likely to be the most appropriate primary outcome measure in a full trial, but that other measures will also be measured as secondary outcomes. We now explicitly note that the systematic review (McLean et al 2012) did not find evidence of benefit for other outcomes on page 5: “A recent systematic review and meta-analysis indicated that self-management DIs may be able to improve asthma control and reduce asthma-related quality of life impairment [12] however there is limited evidence of benefit for other outcomes and larger confirmatory trials are required.”

4. Method: Settings: Are all those General Practices similar one to another? i.e. has a similar characteristics of asthma patients, in the area of remote or urban?

Although there is evidence that asthma outcomes are worse in UK populations from deprived regions (eg Shiue, European Respiratory Journal 2012 40: P3949), the person-based approach to intervention development aims to maximise the reach of interventions such that those from both urban and rural practices benefit. To explicitly address this, we have reworded the method section (p6) which now reads “To ensure we evaluated the intervention across a spread of socio-economic deprivation, practices were purposively selected to be both rural (N = 4) and urban (N = 3), with mean practice deprivation index of 20.60% (SD 10.5); practice socio-economic deprivation deciles = 2, 4, 4, 5, 8, 10, 10, in which lower deciles indicate more deprivation[15]”. We have already addressed some of this issue in the discussion section, in which we discuss the care that must be taken to ensure that digital transformation does not entrench healthcare inequality.

5. Method: Outcomes: It doesn't clear what are the instruments used for measuring Asthma – related QoL? Was it mini AQLQ or AQLQ? Page 6 line 33 mentioned AQLQ, Line 30 page 10, mentioned mini AQLQ (32 items), line 30 page 12 please confirm since Juniper's mini AQLQ had 15 items questions (short version of 32 items in AQLQ)

We apologise for this typographical error – we have now corrected this in the outcomes section to say '15-items' in line with the Mini AQLQ which we used (p10).

6. Method: Outcomes: Regarding the feasibility outcomes, I wonder why outcomes regarding the acceptability of using the DI MBM was measured. Such as whether it designed to be used by all ages of asthmatic patient easily?

We consider ensuring the acceptability of interventions using the person-based approach to be of utmost importance, especially during the development and feasibility trial stage, in order to maximise the effectiveness of interventions during fully powered randomised confirmatory trials. We have amended our description of participant recruitment in the methods section which now reads “ In 46 interviews with 30 patients (purposively selected across a range of age and gender), the intervention was iteratively modified” (p8).

In addition to this, we now explicitly state in our discussion that “We further explored the acceptability of the intervention to people with asthma in a mixed-methods process analysis which will be published separately“ (p23).

7. Results: Page 13 line 54 Primary outcomes : intervention 37/45 and control 43/43 – the less number on the intervention group completed the study?

Yes this is the case. We already explicitly comment on this in the discussion (p21) but to address the reviewer's concerns we have reworded it to read: “Notably, all 8 patients lost-to-follow were in the intervention group. It is possible that patients in the control group were more likely to maintain contact as they were only able to access the intervention upon completion of 12-month follow up measures.”

8. Page 15 line 6. Have any statistical analysis done to see both groups are well-balance and similar?

We did not conduct statistical analysis to confirm the groups are well balanced, as in an RCT balanced groups would be ensured by functional randomisation (which we demonstrated in this trial). In order to account for baseline variance, our primary between-group analysis controlled for baseline

differences. This is in line with current CONSORT guidelines which explicitly state that statistical tests should not be conducted for baseline balance.

9. Page 16 line 45. Are this separately process analysis already published? What it has to be separated? if it could be included in this study.

Including the mixed methods process analysis is unfortunately well beyond the scope of this report due to word count, and would also obscure the main feasibility findings that we aim to focus on here. We consider our secondary measures (anxiety, depression, medication adherence, physiological measures of lung function) and our tertiary measures (intervention usage and engagement) to provide some indication to processes although due to word count we did not conduct detailed analyses. This process analysis is currently being prepared for submission to a peer reviewed journal.

10. Page 17 line 21. (Not statistically significant)...not clear what it means. Is this study expected to be significant statistically? The larger improvement just showed descriptively. I am not sure if the study is power enough to be shown statistically significant.

The study was not powered to have statistical significance, which we acknowledge on page 3 (see response to reviewer 2 comment 3). We have removed the line “not statistically significantly” from the text in line with the reviewer’s suggestion (p17).

11. Discussion: Page 22 line 44: Are both study are similar to compare regarding the intervention? Sample size? And the outcomes measures? If not, stated that the engagement with the intervention was increased is not fully correct.

The RAISIN trial and the MBM trial were similar (FM was an author on both papers) and we have acknowledged the influence of the ‘Living Well with Asthma’ intervention evaluated in the RAISIN trial in the method section (p8): “Pharmacological content was initially based on ‘Living Well with Asthma’, an asthma self-management intervention that previously demonstrated feasibility for self-management[11,19].” As such we consider this statement to be correct.

12. Conclusion: I have not seen clearly which part of the study that saying the MBM is feasible, since the analysis was done descriptively, and there were no significant difference in outcomes measured between Intervention and control group for each outcome.

In line with our primary research objectives, our feasibility outcomes were trial procedures (recruitment strategy, eligibility criteria, consent, withdrawal, randomisation and blinding), intervention usage and engagement, and data analysis techniques. We thank the reviewer for this suggestion. Our discussion now explicitly acknowledge that our findings are in line with our research objectives (p20), reading: “In line with our main research objectives, findings from our randomised controlled feasibility trial demonstrate that a full-size confirmatory trial to confirm effectiveness of MBM, a digital self-management intervention for adults in primary care with asthma is likely to be feasible and acceptable. Our trial procedures, intervention usage and data management were all feasible.”

Comments from Reviewer 2

1. The aim of this study was to assess the feasibility of a trial to evaluate a digital intervention in primary care to improve clinical outcomes of people with asthma, in comparison to usual care. As this is the aim of the study, then any statistical method should be exploratory and it has low power to make any inference about main effects.

We are grateful to the reviewer for their comments and have responded to each point in detail below.

2. Page 2 - line 46: report Interquartile Range (IQR) as well.

We now report the IQR (p2): "36/44 patients in the intervention group engaged with MBM (median of 4 logins, range 0-25, IQR 8)."

3. Page 3 - line 23: A powered study has an ability to identify a true difference between the groups if such difference exist and does not always results in statistical significance results. Therefore remove the sentence "and the study was not powered to show statistical significance" and add if you like the sentence "and the sample size of the study was not determined from a power analysis".

We thank the reviewer for the suggestion and have amended the manuscript by removing the sentence as advised.

4. Page 12 – line 34: All measures were considered continuous. If this is the usual practice and the suggestions of the creators of those measures then it is fine to consider them continuous and apply linear regression models. Otherwise, adjust according to the suggestions of the creators of those measures.

We based our analysis on statistical modelling techniques from usual practice, namely the BREATHE Trial (Bruton et al., 2018) and RAISIN trial (Morrison et al., 2016). As such, we consider linear regression to be appropriate modelling for this feasibility trial for the primary endpoints AQLQ and ACQ. However, we note that we did not conduct regression models for health economic variables (reporting them descriptively) so have removed this. The section now reads: "Exploratory analysis explored group differences in continuous primary endpoint measures (AQLQ, ACQ, HADS, PEI) using linear regression models that controlled for baseline values."

5. Page 12 – line 34: Report all the independent variables included in the multiple linear regression model.

In line with this we have modified our analysis section to explicitly state that our linear regression model controls for baseline variance (p12 – see response to point 4).

6. Page 12 – line 35: It is suggested to use a multilevel repeated measurement model. The highest level is the practices (7 general practices) and the other level is repeated measurements on the individuals over time.

As this trial was a feasibility study, we have not taken this approach as to do so would be a post-hoc analysis that would be underpowered and not in line with our original study protocol. However, we thank the reviewer for their recommendation and will consider this as a primary analysis for a subsequent fully-powered clinical randomized controlled trial.

7. Page 12 – line 41: Define the minimal clinically important difference.

Apologies for this oversight – this has been amended on p12 to read: “Proportions of patients achieving a minimal clinically important difference (MCID) was described for asthma quality of life (the AQLQ MCID is 0.5[16])”.

8. Page 17 – line 41: Correct the 95% Confidence Interval. It should be (-0.22, 0.35). Page 17 – line 45: The same as above.

Apologies for this mistake – this has now been corrected. Please see response to comment 8 for details of specific changes.

9. Table 2 – last column: Are those the results of the difference between the means of the intervention and control group? If they are then the AQLQ is 0.06 lower in the intervention group. Please correct accordingly the Table and the statements in the manuscript.

We apologise for this disparity. We have recoded the values in Table 1 such that they now are now in line with the text (ACQ lower in intervention group indicating better control; AQLQ higher in intervention group indicating better quality of life). This is noted in the methods section (p10) which reads “- Asthma-specific quality of life. Measured using the Mini Asthma Quality of Life Questionnaire (AQLQ[16]), a 15-item 7-point scale in which higher scores represent higher quality of life. - Asthma control. Measured using the Asthma Control Questionnaire (ACQ[23]) , a 7-item 7-point scale in which higher scores indicate worse asthma control.”

The relevant section of the results (p17) now states: “At the 3-month evaluation, patients in the intervention group who completed 3 month follow-up measures (N = 36) had mean improvement in asthma-related quality of life (AQLQ score) of 0.53 (95% CI: 0.31, 0.75), and in the control group of 0.52 (95%CI: 0.30, 0.74), with the between-group difference (controlling for baseline differences) the AQLQ being 0.06 higher (95% CI -0.22, 0.35) in the intervention group, indicating better quality of life. By 12 months, these figures were 0.35 (0.10, 0.60) and 0.21 (-0.09, 0.51) respectively, and the between-group difference had risen to 0.18 (95% CI -0.21, 0.56) higher in the intervention group. In the ACQ analysis, at the 3-month analysis, the between-groups ACQ score was 0.14 lower (95% CI -0.41, 0.13) in the intervention group, indicating better control, and at 12 months was 0.14 lower (95% CI -0.40, 0.11).”

In order to remain consistent, we have applied a similar recoding to all variables in Table 1 (p18) such that the direction of change is the same as that reported in the results section (ie intervention vs. control, rather than control vs. intervention).

10. Page 20 – line 6: Mention the negative binomial model in the Data Analysis section.

We apologize for this omission which has now been reported on p12: “Healthcare utilisation outcomes were explored using a negative binomial model of group count data.”

11. Page 20: Report sensitivity analysis results as you mention it in the Data Analysis

This was conducted by exploring differences in participants lost to follow up with those who remained in the study (p16, L1). We have highlighted that thus was our sensitivity analysis: “Table 1 compares those lost to follow-up to those who remained in the study at 3 months in a sensitivity analysis.”

12. Page 23 – line 21: Power analysis methods should justify the power of the study. Therefore remove the sentence “and so justify a fully powered study”.

Thank you for this suggestion. We apologise for not being clear and have amended this sentence to read “The order of magnitude of the mean between-group improvements in the patient reported measures of control (ACQ) and asthma-related QOL (AQLQ), although not statistically significant with the sample size of this feasibility study, was comparable to that reported in controlled studies of pharmacological[32] and non-pharmacological[8] interventions in asthma, and so justify a confirmatory study with a fully-powered sample.”.

VERSION 2 – REVIEW

REVIEWER	Dr Demetris Lamnisis Department of Health Sciences, European University Cyprus
REVIEW RETURNED	02-Sep-2019
GENERAL COMMENTS	The authors addressed all the issues raised in the previous round of revisions. I believe the manuscript is now adequate for publication